# Printed smart devices for anti-counterfeiting allowing precise identification with household equipment

Junfang Zhang[1,2], Rong Tan[1,3], Yuxin Liu[1,4], Matteo Albino [2], Weinan Zhang[2], Molly M. Stevens [2] & Felix F. Loeffler [1] ✉

Counterfeiting has become a serious global problem, causing worldwide losses and disrupting the normal order of society. Physical unclonable functions are promising hardware-based cryptographic primitives, especially those generated by chemical processes showing a massive challenge-response pair space. However, current chemical-based physical unclonable function devices typically require complex fabrication processes or sophisticated characterization methods with only binary (bit) keys, limiting their practical applications and security properties. Here, we report a flexible laser printing method to synthesize unclonable electronics with high randomness, uniqueness, and repeatability. Hexadecimal resistive keys and binary optical keys can be obtained by the challenge with an ohmmeter and an optical microscope. These readout methods not only make the identification process available to general end users without professional expertise, but also guarantee device complexity and data capacity. An adopted open-source deep learning model guarantees precise identification with high reliability. The electrodes and connection wires are directly printed during laser writing, which allows electronics with different structures to be realized through free design. Meanwhile, the electronics exhibit excellent mechanical and thermal stability. The high physical unclonable function performance and the widely accessible readout methods, together with the flexibility and stability, make this synthesis strategy extremely attractive for practical applications.

With the growing influence of the Internet of Things, information security has never been a more serious concern in modern society. Software-oriented validation systems are currently dominant, but they alone are not able to combat the increasing threats of cyber-attacks[1,2]. Therefore, the development of hardware-based cryptographic primitives[3] is important and urgent to prevent information from being stolen, manipulated, or misused. Physical unclonable functions (PUFs) are one of the most promising hardware-based cryptographic primitives[4–7]. They can generate random and unique bit strings by incorporating stochastic variations during the fabrication process, making them immune to counterfeiting even by the original manufacturer[8–11].

Due to the vast chemical parameter space, chemically generated PUFs can offer a greater encoding capacity and flexibility than the typical silica-based PUFs[12]. Optical PUFs, especially those based on fluorescence properties, have been intensively investigated[13–18].

[1]Max Planck Institute of Colloids and Interfaces, Am Muehlenberg 1, 14476 Potsdam, Germany. [2]Department of Materials, Department of Bioengineering, and Institute of Biomedical Engineering, Imperial College London, London SW7 2AZ, United Kingdom. [3]Soochow University, College of Chemistry, Chemical Engineering and Material Science, Suzhou 215123, China. [4]Freie Universität Berlin, Department of Chemistry and Biochemistry, Arnimallee 22, 14195 Berlin, Germany. ✉e-mail: Felix.Loeffler@mpikg.mpg.de

However, multiple lenses or objectives for fine-tuning the focus are required for information readout, and the long-term stability of the devices is threatened by possible fluorescence quenching. PUF systems based on the readout by surface-enhanced Raman spectroscopy are another major type[19,20], which typically require skilled personnel and time-consuming 2D mapping. Electrochemical PUFs could be more accessible to general end users, but they still require professional equipment for identification[21–24]. In addition, the need for sophisticated preparation processes, toxic semiconductor materials, or expensive noble metals, hinders practical application of these PUF devices[15,16,25].

Here, we print electronic PUFs using an efficient laser-based synthesis process and environment friendly materials for anti-counterfeiting. Our process circumvents the need for sophisticated preparation conditions, toxic materials, and expensive noble metals. The PUF devices can be quickly read out by household ohmmeters without compromising the complexity of the PUF devices (Supplementary Table S1). The observed errors between different measurements, including measurements by different people and different ohmmeters, are negligible. By dividing the data range into small sections that are much larger than the measurement errors, we can obtain 16 bits for each electrode. This greatly improves the security level compared to traditional binary encoding processes. The uniqueness of the PUF devices is achieved by introducing local inhomogeneities during the spin-coating process and a random micro-hole pattern, affecting the connection wires between the electrodes. The hole pattern can be read out using a standard optical microscope, adding an independent method to the resistance readout. Resistive and optical characterization show excellent PUF properties with near-ideal randomness, uniqueness, and reliability. Due to the high flexibility and efficiency of the printing technology, the encoding capacity of the electronic device can be easily increased by printing more electrodes on defined areas. Furthermore, the PUFs are highly stable under mechanical stress and high temperature.

## Results

### Laser-printed PUF electronics

To prepare the PUF devices, the first step is to print a uniform array of polystyrene (PS) spots on the substrate (Fig. 1a). The spots are generated by laser-induced forward transfer (see Methods and Supplementary Fig. S1)[26]. Since an efficient laser absorber is introduced into the printing process, a continuous wave laser at 488 nm is sufficient for material transfer[27]. Then, a hematite precursor film is spin-coated over the spots, causing some spots or spot edges to be randomly washed away. The resulting films are placed in an air oven at 500 °C for 10 min. Remaining PS spots are decomposed at this temperature, resulting in holes of different sizes. Afterwards, electrodes and connection wires are printed by selectively carbonizing a (spin-coated) polymer film on the top of the hematite layer. Since the hematite layer has a good affinity for both polar and non-polar solvents, this polymer film can be flexibly prepared by different materials, including polyvinyl alcohol, polyvinylpyrrolidone, and PS. Meanwhile, we find that the resistance readout can be more stable when we brush the surface of the electronics with colloidal graphite ink (Supplementary Fig. S2). Besides ohmmeters, optical microscopes can be another way for information readout. These two methods are independent of each other, and both are simple and efficient. Manufacturers can use the combination of these two readouts for the data cloud, while general end users can quickly check the resistance code at home.

The thickness map of the electronics (Fig. 1b) shows that the connection wires pass by/over a random number of holes of different sizes. The micro-hole patterns are strongly related to the PS spots and can be flexibly tuned by the laser parameters (Supplementary Fig. S3). Scanning electron microscopy (Fig. 1c) and energy dispersive X-Ray analysis (Fig. 1d) provide a detailed observation of the holes. Carbon is

mainly distributed along the laser writing direction. Both carbon and iron are absent in the hole area, indicating no laser absorption and no carbonization. Therefore, the number and size of the holes can influence the resistance of the connection wires. Furthermore, we also measured the thickness of the hematite layer (Fig. 1e and Supplementary Fig. S4), which shows a significant influence on the resistance. When we double the thickness of hematite layer (from 173.6 nm to 342.0 nm), the resistance of the electrodes decreased almost 10 times (Fig. 1f). In addition, laser parameters (Supplementary Fig. S5), polymer type (Supplementary Figs. S6–8), and carbonization times (Supplementary Fig. S9) can also tune the resistance of the electronics. These factors allow us to tailor the resistance range for practical applications.

### Function of the micro-holes

To investigate the function of micro-holes, we synthesized electronics with ($E_W$) and without ($E_{WO}$) the polymer spot array patterning step, but kept all other parameters constant. The electrode resistances of different devices without micro-holes (Fig. 2a) show a randomness caused by the inhomogeneity (i.e., thickness variation) of the spin-coated hematite film (Fig. 2f left). Nevertheless, a clear trend can be observed in $E_{WO1}$ to $E_{WO3}$ (Fig. 2a, c), suggesting that an inhomogeneous hematite layer alone (Fig. 2f center) is not sufficient for the required high randomness in the electrode resistance. For the electronics with micro-holes (Fig. 2f right), this trend disappeared, confirming that the micro-holes are critical for a high randomness (Fig. 2b, d). Comparing more devices (Fig. 2e, $n = 16$), it becomes clear that micro-holes efficiently increase the accessible resistance range, which is beneficial for the PUF performance. Devices with micro-holes show $0.062 \pm 0.088$ and $0.902 \pm 0.124$ for intra- and inter-device correlation (Supplementary Fig. S10), respectively, suggesting excellent uniqueness and robust responses to repeated challenges. In contrast, devices without the micro-holes (Supplementary Fig. S10) show a smaller inter- and larger intra-correlation, representing a lower PUF performance. Their inter- and intra-correlation functions overlap, making reliable identification often impossible. Meanwhile, we noticed that the resistance of $E_W$ devices was typically higher for electrodes at the diagonal positions (Fig. 2b, d, electrode #1) and lower for those in the horizontal direction (Fig. 2b, d, electrode #4). We assumed that the laser writing direction of the connection wires also plays an important role in the resistance of the electrodes. Therefore, we investigated the impact of a uniform writing orientation (Fig. 2g1) and a defined orientation (90° and 45°) in reference to the connection wire direction (Fig. 2g2,g3) on the resistance distribution (Supplementary Fig. S11). Meanwhile, using a 45° laser writing direction in reference to the connection wires eliminates the observed trend in the diagonally positioned electrodes. Concluding, a 45° laser writing orientation (Fig. 2g3) together with micro-hole structures result in optimum randomness of the electronics. These parameters were selected for the following experiments.

### Characterization of PUF properties

To characterize the micro-hole structure, the PUF electronic device was scanned by an optical microscope with 5× magnification (Fig. 3a, b). The obtained pictures were converted into binary color images. By adjusting the contrast threshold, the black connection wires and electrodes can be made invisible for subsequent image segmentation. After image extraction of the micro-holes, we analyzed their detailed features and PUF performance. The hole diameter (Fig. 3d) ranges between ~10–40 μm and follows a Gaussian distribution. From the height map of the micro-holes (Fig. 3c), three layers (blue = insulating, orange = slightly conductive, brown = conductive) can be distinguished by the software (see Methods), which automatically introduced two thresholds according to the histogram distribution. These height layers of the hematite surface with holes correlate with the electric resistance of the laser carbonized wires, as discussed previously (Fig. 1f). The insulating layer has a height below about 200 nm,

suggesting it is the bare glass substrate at the bottom. Glass does not absorb the laser radiation at 488 nm. Thus, no carbonization occurs on the micro-holes, which makes this layer insulating. The orange layer with a hematite thickness below ~600 nm can be found at small holes and at the edges of large holes, yielding carbon materials with high resistance. The brown layer represents thick hematite layers (above ~600 nm), which give a more efficient carbonization process and are the most conductive layer within the electronics.

For statistical analyses, we generated 100 unique PUF electronics using the same laser parameters (Appendix in Supplementary Information). Each sample was scanned twice with repositioning of the sample in between. A simple hole position analysis (e.g., hole position present/not present) would make the optical PUF label easily clonable, since counterfeiters could duplicate the pattern with the same hole positions through a deterministic printing process. To utilize the high randomness of the optical pattern, fine level recognition is required. Therefore, the open-source LoFTR (Detector-Free Local Feature

Matching with Transformers)[28] algorithm was adopted for the micro-hole pattern identification process. LoFTR can achieve pixel-wise dense matches at a fine level with a maximum match number of 4256. Comparing images from the same PUF device (two independent image acquisitions, sample repositioned in between), LoFTR could typically match almost all features (Fig. 3e top). When comparing different PUF devices, only a small number of matches can be observed (Fig. 3e bottom). The similarities of 100 PUF electronics were calculated by the match ratios with respect to the maximum number of matches (Fig. 3f). There is a significant difference between the data on the diagonal (representing the similarities of two independent scans of the same PUF device) and the rest (similarities of different PUFs) in the heat map. This guarantees high reliability of the identification process.

Instead of being randomly distributed around 1, almost all intra-correlations are exactly 1 (Supplementary Fig. S12), indicating an extraordinarily high repeatability. By calculating the false authentication (FA) and authentication error (AE) as a function of the decision

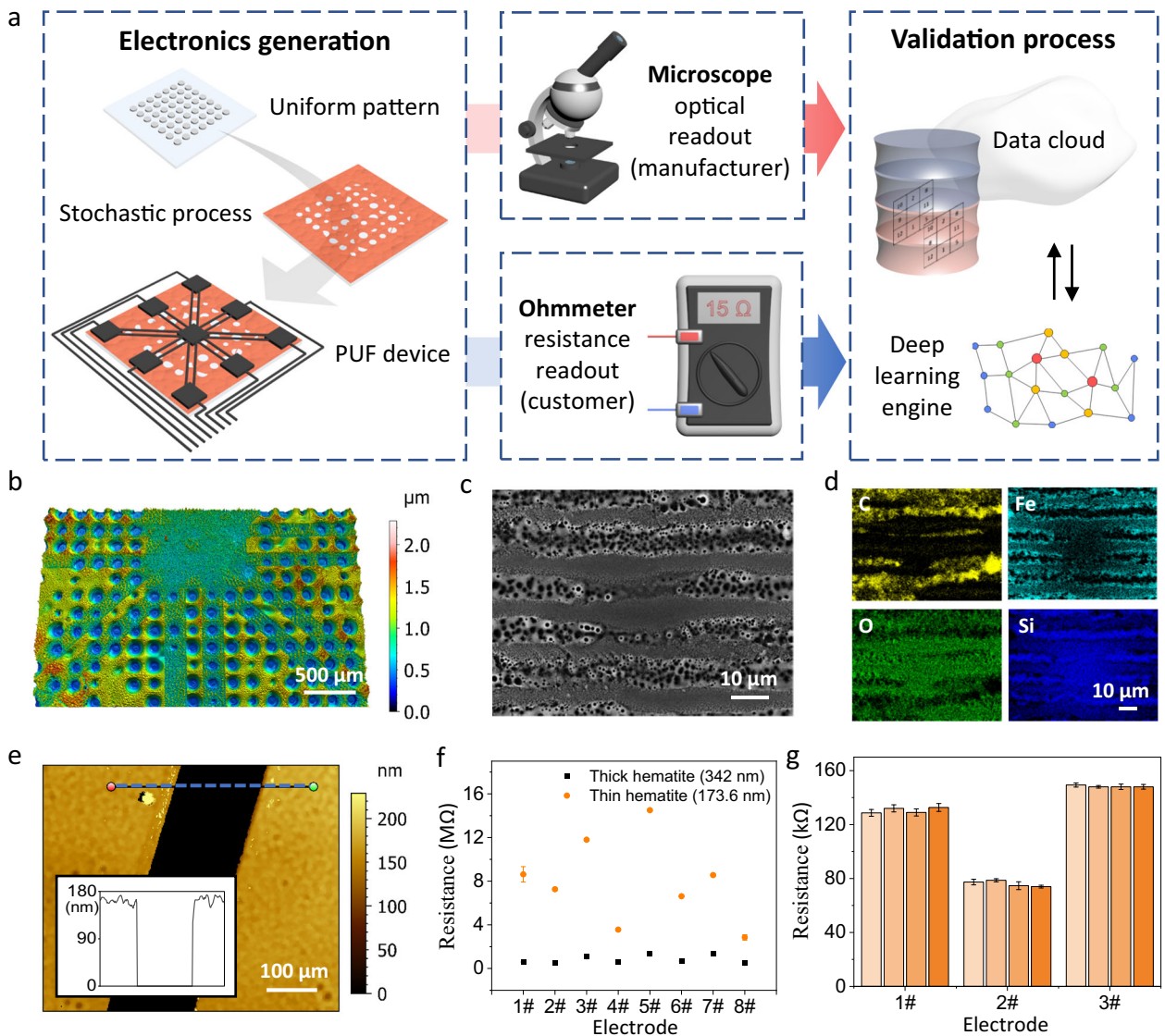

**Fig. 1 | Synthesis and characterization of PUF electronics. a** The PUF electronics are synthesized by spin coating and laser printing. Precise identification can be realized by an optical microscope and ohmmeter with a deep learning algorithm. **b** Height map of the electronic device. **c** Scanning electron microscopy imaging of the connection lines between electrodes, passing through a hole. **d** Elemental distribution map by energy dispersive x-ray analysis. **e** Height map and profile of the hematite layer. **f** The resistance of the electrodes is highly dependent on the thickness of the hematite layer, shown as average of $n = 3$ measurements with standard deviation. **g** Resistance measured by four different individuals with two ohmmeters from different brands, shown as average of $n = 3$ measurements with standard deviation.

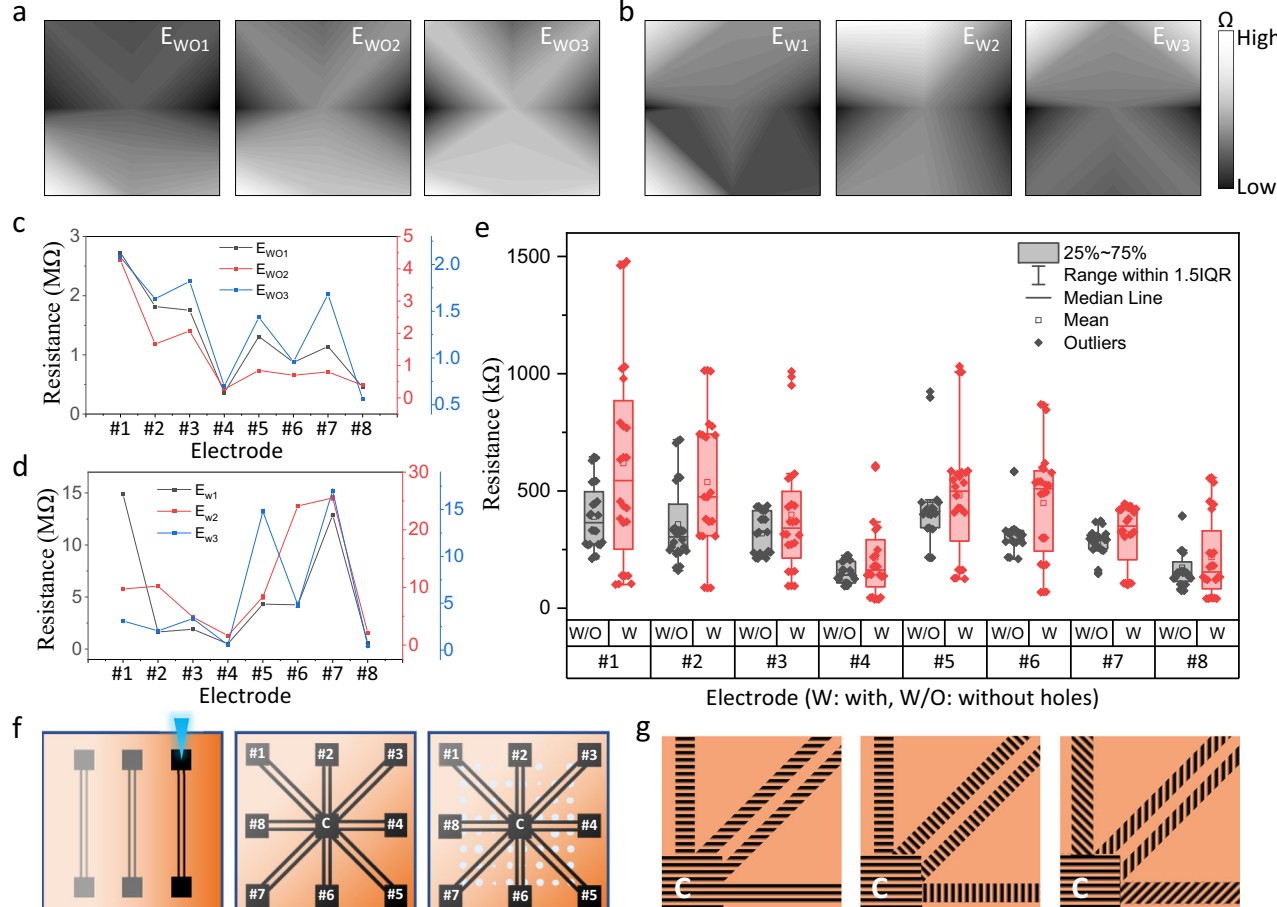

**Fig. 2 | Influence of the micro-holes on resistance.** Resistance heatmaps (3×3 matrix) of the electrodes of three devices (**a**) without micro-holes ($E_{WO}$) by skipping the polymer patterning and (**b**) with micro-holes ($E_W$). Electrode resistances of the same devices without (**c**) and with (**d**) micro-holes. Each device was measured three times, showing mean value and standard error. **e** Investigation of the resistance range ($n = 16$ devices), 8 without (W/O, black) and 8 with (W, red) micro-holes. **f** Local inhomogeneities of the hematite layer (thickness variations indicated by gradient from light orange to dark orange) result in different laser carbonization

degrees (light to dark black) of the polymer (left). This causes a randomness in the resistances for the electronics without micro-holes (middle). The electronic with holes gains its randomness from both the inhomogeneous hematite layer and micro-hole structures, significantly increasing the resistance range and randomness (right). Electrodes are numbered, C is the central electrode. **g** Magnification of the top right area of the central electrode C showing different laser writing directions for carbonization (left: 0°, all one direction; center: 90° in respect to connection wire direction; right: 45° in respect to connection wire direction).

threshold, the LoFTR-based probability of cloning can be estimated to be below $10^{-11}$ at a decision threshold of 0.43. In addition, we also calculated the Hamming distance (HD) to evaluate the PUF properties (Supplementary Table S1). The average bit uniformity is 0.481 with a narrow standard deviation (SD) of 0.056. This is close to the ideal value of 0.5, confirming high randomness. The calculated uniqueness and reliability of the PUF devices suggest that the micro-hole patterns significantly differ from each other and provide robust responses to repeated challenges.

The electrode in the center served as the counter electrode. The resistance between the outer eight (#1–#8) electrodes and the counter electrode is read out by an ohmmeter. All 100 PUF devices were measured twice and the interquartile range (IQR) was calculated. Based on the outlier formula, values greater than 1.5x IQR were considered as outliers and defined as 0. Values within the 1.5 IQR were divided into 15 intervals (1–15, Fig. 4a). Since the standard error for measurements from the same sample is very small (Fig. 1g), more intervals should be possible if required. After encoding, the resulting hexadecimal strings were used to characterize the PUF resistance. The inter- and intra-device correlations were 0.937 ± 0.071 and 0.029 ± 0.053, respectively (Fig. 4b) and they follow a Gaussian

distribution. The uniqueness between any two PUF patterns is generally defined as:

$$\text{uniqueness} = \frac{2}{k(k-1)} \sum_{i=1}^{k-1} \sum_{j=i+1}^{k} \frac{\text{HD}(R_i(n), R_j(n))}{n} \quad (1)$$

where $R_i(n)$ and $R_j(n)$ are the $n$-value responses of the $i^{\text{th}}$ and $j^{\text{th}}$ PUF patterns, respectively, and $k$ is the total number of the PUF patterns. Based on the equation, the ideal value of uniqueness is 1/2 for binary keys, 2/3 for ternary keys[29,30], etc., and 15/16 for hexadecimal keys. The inter-device uniqueness for the resistivity readout is 0.937 ± 0.071, which is close to the ideal value of 15/16 = 0.9375, giving solid evidence for a high performance. At decision thresholds of 0.48, the probability of cloning can be estimated to be below $10^{-9}$ (Fig. 4c). These results indicate high randomness, uniqueness, and reliability of the electronic PUFs. Notably, the hexadecimal strings significantly improve the theoretical key space in comparison to the commonly used binary keys (Supplementary Table S2). The electronic PUFs promise a simple and fast readout that is accessible to general end users, without compromising the performance and data capacity of the PUF.

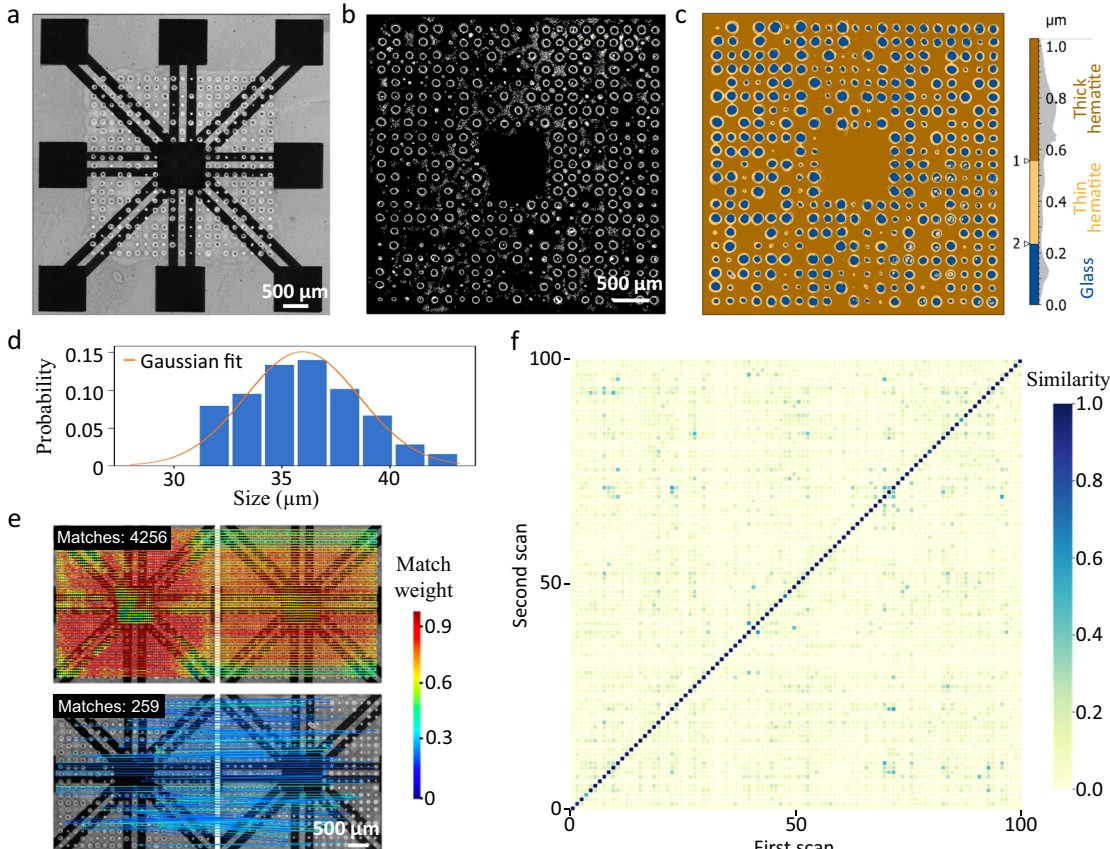

**Fig. 3 | Characterization of the micro-hole structure.** Optical microscopy images of (**a**) the PUF electronic device and (**b**) the micro-holes (magnification: 5x). **c** Height map of the micro-hole structures. **d** Size distribution of the holes. **e** Feature matching by the LoFTR algorithm for the same (top) and two different (bottom) PUF electronics from two independent scans. **f** Heat map of WLI similarity values (match ratio, referenced to maximum measured match number 4256). 100 unique PUF electronics, created using the same laser parameters, were used for the analysis in **d** and **f**.

Furthermore, the micro-hole patterns and electronic resistances are highly independent readout methods. The combination of these methods can greatly increase the security level of PUF devices and makes them extremely difficult to counterfeit.

## Stability of PUF devices and design flexibility

The synthesis process for PUF devices is highly time- and cost-efficient, allowing it to be applied to valuable products, such as artworks. The straightforward readout methods give two independent keys, a binary array and a hexadecimal string, which are sufficient for high-security levels (Fig. 5a). For practical applications, device stability under different environments is critical. Therefore, the surface of the electrodes and resistance values were recorded before and after mechanical stress (Fig. 5b) and high-temperature treatment (Fig. 5c). For mechanical stability, we stick and remove tape on the device, repeating five times with fresh tape each turn. Scanning the surface of the device by white light interferometry revealed no significant differences (Supplementary Fig. S13) and the resistance values remained unchanged. Both, before and after treatment, each electrode was measured three times with negligible standard errors. For the thermal treatment, the device was placed in an air oven at 200 °C for 1 h. No changes on the electrode surface and in the resistance values were observed, proving the thermal stability of the device. Moreover, since the PUF electronics are directly printed by laser, the configuration of the electronics can be flexibly designed with more electrodes and different connection wires (Fig. 5d). Notably, when we increased the electrode number from 8 to 32, the encoding capacity will greatly increase from

$8^{16}$ to $32^{16}$ ($\approx 10^{24}$), which is much larger than the current world population.

## Discussion

In conclusion, we proposed an efficient synthesis method for printable PUF electronics. By spin-coating an iron precursor solution on the surface of a polystyrene spot pattern, the spots are randomly washed away. In the subsequent annealing step, the iron precursor is decomposed into a hematite nanolayer, while the remaining polystyrene spots are burned to form (empty) micro-hole patterns in the nanofilm. Since hematite can efficiently absorb laser light and convert it into heat, the carbonization of another polymer layer on top can be realized under controlled laser irradiation. Based on this, electrodes and connection wires were written by laser to generate the electronics. Since the micro-holes do not absorb the laser, the polymer is not carbonized. The number and size of micro-holes passed by each connection wire are random. According to the PUF device characterization, their resistances exhibit near-ideal randomness, uniqueness, and reliability. Notably, the resistances of the electrodes can be easily read out by an ohmmeter, allowing identification for general end users even at home. The standard deviations of the readout are negligible, therefore, it allows hexadecimal strings encoded by the electronics, offering very large encoding capacities. Meanwhile, micro-hole patterns can be recorded by normal optical microscopes without special requirements on objective lenses and incident angles. An open-source deep-learning algorithm is introduced for precise feature extraction and comparison during the identification. The combination of resistance and optical

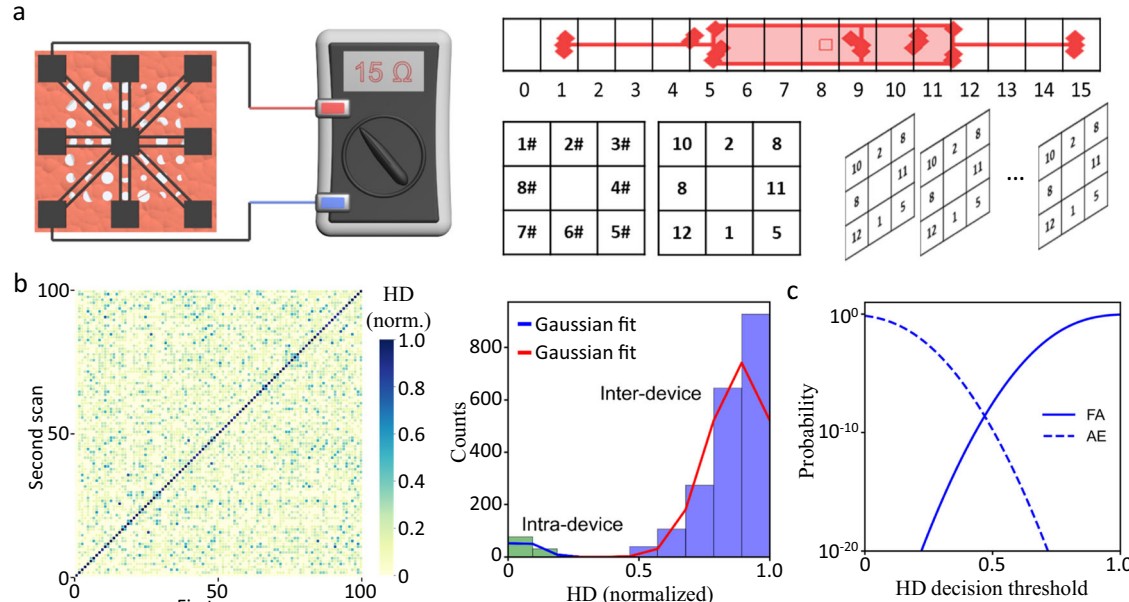

**Fig. 4 | Electronic resistance characterization of PUF devices. a** The resistance of the electrodes can be simply read out with a handheld ohmmeter. The range of the resistance values for each electrode can be divided into 16 intervals (example data from Fig. 2e, $E_W$ electrode #2). The resistance value is encoded accordingly. **b** Heat map of intra-device (diagonal) and inter-device Hamming distance (HD) obtained from 100 unique PUF patterns generated using the same laser parameters. Device uniqueness of the PUF patterns was characterized by inter-device HD. The readout reproducibility of the PUF patterns was characterized by the intra-device HD, where each PUF pattern was scanned/measured twice. **c** Cumulative distribution functions, showing the probabilities of false authentication (FA) and authentication error (AE) as a function of the decision threshold.

readout methods drastically enhances the security level of these PUF devices. Similar to other reported PUF device fabrication methods, our process involves several processing steps. However, they are simple and circumvent the problems in PUF material pre-fabrication that typically involve sophisticated synthesis conditions, toxic materials, or expensive noble metals. Due to the flexibility of our printing process, the configuration of the connection wires and the number of electrodes can be freely designed, which increases the adaptability of the devices to different practical applications. In addition, the electronics show excellent mechanical and thermal stability, proving their potential even in extreme environments.

## Methods
### Synthesis of PUF electronics
Step 1: Laser-induced forward transfer. 50 mg PS (av. Mw = 35,000, Sigma) is dissolved in 500 µl dichloromethane. The solution is spin-coated on the surface of a hematite donor slide at a rotation speed of 70 rpm. More details on the development hematite donors can be found in our previous work[27]. The hematite donors are prepared as follows: 125 mg of PVA (polyvinyl alcohol, av. Mw = 9000–10,000, Sigma) and 125 mg of $Fe(NO_3)_3 \cdot 9H_2O$ (98 %, Acros) are dissolved in 250 µl of $ddH_2O$. 250 mg of PEG (polyethylene glycol, av. Mw ≈ 20,000, Sigma) and 250 mg of $Fe(NO_3)_3 \cdot 9H_2O$ are dissolved in 250 µl of methanol. The two solutions are mixed by stirring for 20 min, sonicated for another 20 min to remove air bubbles, and spin-coated on a glass substrate (70 rpm), which is annealed at 500 °C in an air oven for 3 h. This hematite donor slide, spin-coated with a PS layer, is placed in the laser-induced forward transfer (LIFT) setup on top of a glass acceptor slide (future PUF device). With 65% power and 500 µs irradiation time, PS spot arrays are generated on the glass acceptor slide. Step 2: 1 g $Fe(NO_3)_3 \cdot 9H_2O$ (98%, Acros) and 1.5 g PEG (Polyethylene glycol, av. Mw ≈ 20,000, Sigma) are dissolved in 1 ml distilled water. Then, 500 µl of the solution is spin-coated on the PS spot array on glass, obtained in the previous step. Sometimes, we observed that the spin-coated film shows a fast dewetting at the edges. This did not influence the behavior

of the electronics, since the edge area was not used. To prevent the dewetting, 5 µL polyvinyl alcohol in water (0.5 g/L) was added to the solution as an additive for the spin-coating to stabilize the edges. The spin-coated film was annealed in an air oven at 500 °C for 10 min. Step 3: Another polymer layer (50 mg in 500 µl solvent) is spin-coated onto the annealed film. Then, the LIFT system is used to print the electrodes and connection wires by carbonizing the polymer. Repeating step 3 once more or brushing the surface with colloidal graphite ink can help to stabilize the measured resistances of the PUF device.

### Characterization of PUF electronics
We designed and built a semi-automatic resistance measurement setup (Supplementary Fig. S14). It allows simple positioning, electrode contacting, and rapid measurement with a rotary selector switch. The thickness map of the electronic PUF device was acquired with a smartWLI compact (Gesellschaft für Bild- und Signalverarbeitung (GBS) mbH, Illmenau, Germany) with low magnification (5× Nikon CF IC Epi Plan DI−Mirau). Further analysis was performed in Mountains-Map 8.0 with the threshold detection method and the step method, respectively. For resistance heatmaps, the resistance of each of the eight electrodes of a PUF device was measured three times, the average was calculated, and arranged in a 3×3 matrix. Then, the median resistance of the eight electrodes was calculated and placed in the center of the 3×3 matrix. Based on this 3×3 matrix, we generated a 2D contour map in OriginPro 2022. The display mode was set as a grayscale color palette without contour lines.

To evaluate the PUF characteristics, the data were transformed into binary signals by defining the median value as a threshold for the analysis of the optical readout. The bit uniformity estimates the distribution of logic-0 and logic-1 in a given set of PUF responses and can be calculated with the following equation:

$$\text{bit uniformity} = \frac{1}{k} \sum_{i=1}^{k} R_i \quad (2)$$

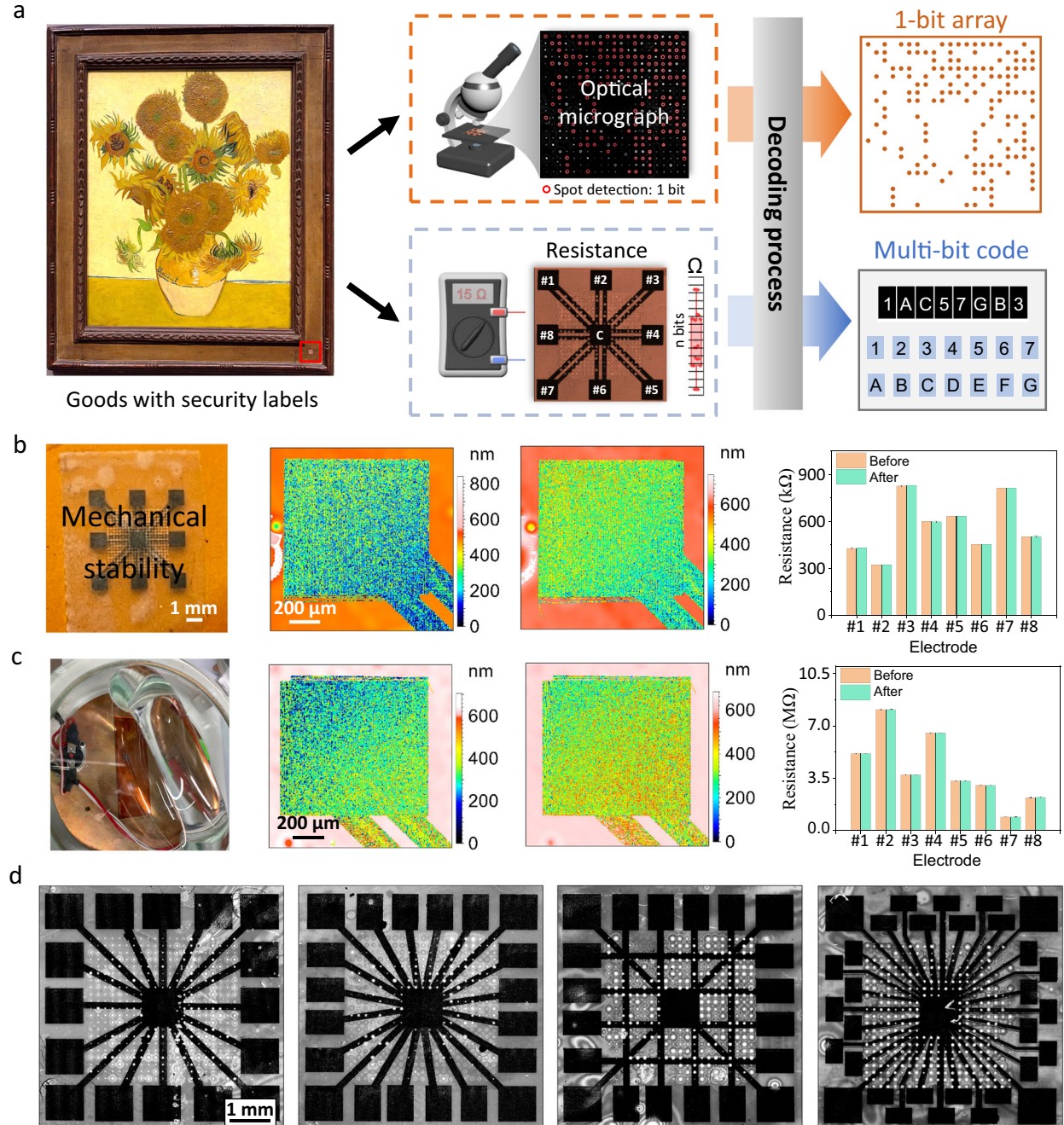

**Fig. 5 | Application and stability. a** The PUF device can be attached to valuable products (Sunflowers by Vincent van Gogh, National Gallery, London, https://commons.wikimedia.org/wiki/File:Sunflowers_National_Gallery.jpg, public domain). Optical microscopes and ohmmeters can be used for the identification process. **b** Mechanical (5x tape attachment) and **c** thermal stability (200 °C, 1 h) test of the device. The surface topography and resistances (shown as average of $n = 3$ measurements with standard deviation) are recorded before and after the test. **d** Variants of electronics bearing more electrodes or different shapes of connection wires for more sophisticated PUF requirements.

where $R_i$ is the $i^{th}$ bit of the PUF pattern and $k$ is the total number of PUF patterns.

All PUF responses were read out or scanned twice and the reliability was evaluated with the following equation:

$$\text{reliability} = 1 - \frac{1}{k} \sum_{i=1}^{k} \frac{1}{T} \sum_{l=0}^{T} \frac{\text{HD}(R_i^0(n), R_i^l(n))}{n} \quad (3)$$

where $R_i^l(n)$ is the $n$-bit response from the $i^{th}$ PUF at the $l^{th}$ trial, $T$ is the number of trials and $k$ is the total number of PUF patterns.

## LoFTR algorithm

The LoFTR algorithm[28] is a framework for image feature matching. Instead of sequentially detecting, describing, and matching image features, it first establishes a pixel-wise dense match and then refines the matches. Unlike traditional methods that use a cost volume to search for corresponding matches, the framework applies self- and cross-attention layers from its transformer model to obtain feature descriptors on both images. The global receptive field provided by the transformer model enables the LoFTR algorithm to produce dense matches, even in low-texture areas, where traditional feature detectors

typically fail to produce repeatable points of interest. In addition, the framework model is pre-trained on indoor and outdoor datasets to recognize the type of image being analyzed, with features such as self-attention. The LoFTR module uses self- and cross-attention layers in transformers to transform the local features to be context- and position-dependent, which is crucial for LoFTR to obtain high-quality matches on indistinctive regions with low-texture or repetitive patterns. For image pair ($I^A$, $I^B$), their similarity is defined by the number of matched features:

$$\text{similarity}(I^A, I^B) = \frac{N_{\text{matches}}(I^A, I^B)}{N_{\text{features}}} \qquad (4)$$

where $N_{\text{matches}}(I^A, I^B)$ is the number of matching features and $N_{features}$ is the number of extracted features. The LoFTR algorithm extracts 4800 features in total, however, for the border area, a mask has been applied for practical analysis. Therefore, the maximum number of possible extracted features is 4256. The resulting colormap shows the matching probability $P_c$. In the case of dual-softmax, it is obtained by

$$P_c(i,j) = \text{softmax}(S(i,\cdot))_j \cdot \text{softmax}(S(\cdot,j))_i \qquad (5)$$

The score matrix S between the transformed features is calculated by

$$S(i,j) = \frac{1}{\tau} \cdot \left\langle \widetilde{F}^A_{\text{tr}}(i) \widetilde{F}^B_{\text{tr}}(j) \right\rangle \qquad (6)$$

where $\widetilde{F}^A_{\text{tr}}$ and $\widetilde{F}^B_{\text{tr}}$ are the added features, which enter the LoFTR module for processing.

## Data availability
All data are available in the manuscript or the supplementary materials.

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

## Acknowledgements

The authors thank Guangkai Li for the technical support with machine learning. This work was supported by the funding from the German Federal Ministry of Education and Research (BMBF, grant number 13XP5050A, F.F.L.), the China Scholarship Council (J.Z.), and the MPG-FhG cooperation (Glyco3Display, F.F.L.). M.M.S. acknowledges support from the Royal Academy of Engineering Chair in Emerging Technologies

award (CiET2021\94). M.A. acknowledges support from the Department of Materials and I-X at Imperial College London. The authors thank the Max Planck Society for financial support.

## Author contributions

J.Z. conceived the idea, designed and performed the experiments. R.T. supported the experiments. Y.L. contributed to the analysis and design of the figures. W.Z. assisted the data collection for statistical analysis. M.A. assisted in the material characterization during revision. M.M.S. supported the project including funding support. F.F.L. supervised the project. J.Z. and F.F.L. wrote and revised the manuscript.

## Funding

## Competing interests

The authors declare no competing interests.
