## [Peer Review File · Nature Communications]

REVIEWER COMMENTS

Reviewer #1 (Remarks to the Author):

The authors present the fabrication of PUF electronics by their previously developed laser printing method. Specifically, they achieved 1) PUF readout by household equipment, including optical microscope with low magnification (5×) and ohmmeter, 2) high resistance randomness for anti-counterfeiting, 3) Well reproducibility and uniqueness, and 4) good mechanical and thermal stability. Overall, it is an interesting and novel work. However, this manuscript needs to solve the following concerned issues before it can be considered for publication.

Q1. The authors claimed that their fabrication method of PUF label is facile. However, compared with the existing main fabrication methods (e.g., drop-casting [Nat. Commun. 11, 516 (2020)], inkjet printing [Nat. Commun. 2019, 10, (1), 2409], etc.), it is hard to believe the fabrication method in this paper is simple and facile. Instead, it is rather complicated comprising several processes: 1) spin-coating of polymer, 2) printing polymer spots on substrate by laser-induced transfer, 3) spin-coating of hematite layer, 4) thermal annealing at high temperature, 5) printing electrodes and connection wires, and 6) brushing the surface of the electronics with colloidal graphite ink. Please explain why the fabrication method in this paper is facile.

Q2. The optical encoding demonstrated in Figure 5a is not unclonable, because it uses the optical pattern (spot detection) for anticounterfeiting. Although each spot is unclonable (size, shape, etc.), the whole optical pattern is not unclonable if just detecting the presence of spot. Because the pattern is fabricated by the deterministic laser printing process, which could be easily copied by counterfeiters if they know the optical patterns.

Q3. The electronic resistance encoding in this paper is just 8-digit (or 32-digit) long hexadecimal coding, which is still relatively lower than some hexadecimal optical coding with 50×50 digit long (or larger) within only hundreds of micrometers detection range. Therefore, the capacity of resistance encoding should be improved.

Q4. It seems time-consuming and inconvenient to conduct electronic resistance measurement for encoding. As it needs to connect the electrodes of ohmmeter with each specific electrode in such small PUF labels to collect each datapoint, which means that at least 8 measurements are needed for one label. Therefore, the time consumption, difficulty of operation, misoperation, etc. should be considered for practical applications.

Q5. The PUF encoding area/spot is relatively large, which is easy to be copied compared with nano- or micro- PUFs.

Q6. Figure 2f is not clear, the degree of darkness is representing the resistances variations? Or? And the other two (middle and right) figures of 2f are not mentioned in the main text.

Q7. How Figure 2a and 2b are obtained should be explained in Methods.

Q8. From the height map of the micro-holes (Figure 3c), three layers (insulating, slightly conductive, conductive) can be distinguished. However, in Figure 1f, the resistance of thin layer electrode has a large range. So, is the resistance map from the height map trustworthy in Figure 3c?

Q9 Some quantitative analysis of the height maps shown in Figure 5b and 5c should be conducted. As it is hard to tell from the figures that there are no significant differences before and after treatment.

Reviewer #2 (Remarks to the Author):

This manuscript reported the printed PUFs having both resistive and optical keys. Authors claimed that their dual key PUFs offer high complexity and data capacity, which can be easily read out using common equipment such as an ohmmeter and an optical microscope.

While I find their concept interesting, I have reservations about their main argument. The authors claim that the combination of resistivity and optical readouts results in significantly higher complexity and randomness. However, based on the provided data, the resistivity readout does not exhibit sufficient complexity as a PUF; in fact, it performs poorly.

The inter-device uniqueness for the resistivity readout, measured using an ohmmeter, is reported as 0.937 ± 0.071 . Although the authors insist that this value indicates high randomness, it deviates significantly from the ideal value of 0.5. Contrary to their claim, this result demonstrates that the resistivity readout offers such poor randomness that it is challenging to use it as a PUF. On the other hand, only the optical readout provides a reasonable inter-device uniqueness value (0.454 ± 0.057), which, however, is not even as high as that of numerous other previous optical PUFs.

Furthermore, the authors argue that the resistivity range of 0 to 1500 k Ω can be divided into 16 levels, allowing for 16-bit operations. For this claim to hold true, all levels should be evenly addressed with the same probability. However, the resistivity measurements depicted in Figure 2c, d, e, and Figure 4a reveal a relatively ununiform distribution, with some resistance ranges (e.g., 1000-1500 k Ω) lacking any hits.

Considering these points, the resistivity readout appears unsuitable for use in PUFs. While their printed device still functions as an optical PUF, I fail to see any significant excellence or advancement compared to previously developed optical PUFs.

Overall, I do not believe this work qualifies for publication in the renowned journal Nature Communications.

Minor comments:

1. Figure 5c displays resistance in the M Ω scale, while Figure 5b shows resistance in the k Ω scale. Is this a typographical error, or does the resistance actually vary to such an extent from sample to sample?
2. Since resistance varies with the extent of oxidation, I recommend conducting a stability test against oxidation for an extended period or performing an accelerated-oxidation experiment under controlled conditions.
3. Figure S10: The x-axis sampling space of the left-right histogram differs. It should be the same for comparison purposes.
4. Figure 3: How did you obtain the height information from the conventional optical image shown in Figure 3a, and how did you identify the insulating, slightly conductive, and conductive regions based on that information? I couldn't find reasonable explanations in the manuscript or the supplementary information. Additionally, what is the conductivity range for each of the three regions?
5. Please provide the electrical conductivities for the pure carbonized-layer at 45° and 90° angles.
6. Please include XPS and Raman data for the carbonized-layer and provide an explanation in the text.
7. It would be helpful to label the electrode numbers in Figure 2f to aid reader understanding.
8. Hematite is an optically interesting material with a conductivity of approximately 10-11 S/cm at room temperature. If the electrical conductivity of hematite is improved, what is the expected performance of this operation?
9. In Figure 2g, the image of the carbonized-layer exhibits periodic light and dark patterns indicating different conductivities. This could have a significant impact on ohmmeter characterization and may negatively affect reproducibility. Could the authors explain the rationale behind this approach?
10. Contrary to the description on Page 5, line 144, where electrodes #1, #3, #5, and #7 in diagonal positions are said to have high resistance, the resistance of #3 electrode does not appear to be large. Could the authors provide clarification on this inconsistency?

Reviewer #3 (Remarks to the Author):

This work proposes a method to create a PUF for an anti-counterfeiting purpose. While its novelty requires justification (its novelty is ambiguous so far), the method itself is technically sound. Below concerns need to be taken into account. I recommend a major revision to address the following points.

1. Firstly, this approach is suitable for device identification, not authentication. The authors can gain a detailed understanding between PUF identification and authentication by referring to [8]. Re-read [8] very carefully.
2. Using multiple characteristics e.g., both resistance and optical parameters concurrently to create a PUF has been investigated in the literature. There is a lack of comparison with existing closely-related works. Performing this comparison thoroughly would help to support novelty of your work.
3. Line 22 does not seem to be appropriate for the abstract, as all the later descriptions have nothing with securing the AI.
4. Line 25 '2-bit strings' is an incorrect claim without specific condition. For example, SRAM PUF can indeed generate hundreds of thousands of bits.
5. Line 55-58. The comments about electronics PUF's practicality is incorrect. Note that PUF has been commercialized and its main type PUF is SRAM PUF, which is indeed practical. This SRAM PUF does not need any extra hardware for authentication.
6. Line 198, usually intra-distance (equals to bit error rate given digitized PUF response) should be approaching to 0 not 1 to indicate high reliability.
7. Line 201, is this identification capability is estimated by a 16-bit response? Usually, with same inter-distance (randomness) and intra-distance (reliability), increasing the PUF response length can improve the identification capability.
8. Figure 5 a, what if the PUF is cut out and placed on the other painting's frame to perform substitution attack?
9. Reliability evaluations usually need to be performed under varying operating conditions. What if the temperature (e.g., room temperature) of the reference/enrolled response is different from the temperature (e.g., 0) of regenerated response?
10. Can the optical PUF property be characterized with inter-HD and intra-HD in a similar way to Fig4?

11. The adoption of a machine learning model needs more elaboration to detail its role and specific operation.

12. Please carefully proof-read the manuscript for English language and expression. For example, the word “facile” is not a good word for a scientific paper. It is one of those strange English words that can both be a compliment and an insult depending on how it is said.....so is best avoided in a journal article.

13. All single letter math variable like n must be in italic. All multiple letter variables, labels, and functions must NOT be in italic, For example, in your paper: softmax, matches, HD, ...etc must not be in italic.

14. In math, when you have multiple letter names, do no mix cases. It does not look good. So for example: Similarity  similarity

REVIEWER COMMENTS

Reviewer #1 (Remarks to the Author):

The authors present the fabrication of PUF electronics by their previously developed laser printing method. Specifically, they achieved 1) PUF readout by household equipment, including optical microscope with low magnification (5×) and ohmmeter, 2) high resistance randomness for anti-counterfeiting, 3) Well reproducibility and uniqueness, and 4) good mechanical and thermal stability. Overall, it is an interesting and novel work. However, this manuscript needs to solve the following concerned issues before it can be considered for publication.

Q1. The authors claimed that their fabrication method of PUF label is facile. However, compared with the existing main fabrication methods (e.g., drop-casting [Nat. Commun. 11, 516 (2020)], inkjet printing [Nat. Commun. 2019, 10, (1), 2409], etc.), it is hard to believe the fabrication method in this paper is simple and facile. Instead, it is rather complicated comprising several processes: 1) spin-coating of polymer, 2) printing polymer spots on substrate by laser-induced transfer, 3) spin-coating of hematite layer, 4) thermal annealing at high temperature, 5) printing electrodes and connection wires, and 6) brushing the surface of the electronics with colloidal graphite ink. Please explain why the fabrication method in this paper is facile.

R: We understand the concern, but our process not only includes the printing of the PUF devices, but also all the synthesis steps of the functional materials. For the mentioned processes, whether drop-casting or inkjet printing, they are only the final step in the production of the PUF device. Before that, the fabrication requires multiple steps that are low in yield or involve difficult conditions:

1. For the generation of Raman PUF labels via drop casting (Nat. Commun. 2020, 11, 516), these steps need to be followed: 1) synthesize Au cores; 2) reduce CTAC concentration by washing and centrifugation; 3) introduce Raman reporters under ultra-sonication; 4) incubate the solution and perform multiple washing steps; 5) rapidly mix ascorbic acid, modified Au cores, CTAC and HAuCl_4 under vigorous sonication; 6) prepare a silica substrate with metallic marks; 7) drop cast the solution and dry it for Raman measurements.

2. In the second example, using inkjet printing (Nat. Commun. 2019, 10, 2409), it first required the synthesis of core-shell quantum dots to generate fluorescent anti-counterfeiting labels. Therefore, the following steps had to be performed: 1) mixing CdO, oleic acid, and 1-octadecene; 2) heating it to 160 °C and degassing for 15 min; 3) heating to 310 °C under N_2 gas protection; 4) injecting Se precursor; 5) cooling the solution to 300 °C and remaining for 5 min; 6) growing CdS shell (injecting S source, reacting with cadmium ions for 20 min); 7) growing CdZnS shell, 8) purification steps. Afterwards, the corresponding inks need to be prepared and the substrates have to be specifically pre-treated.

In contrast, for our approach, all steps are quite simple and straightforward. A compatible laser setup can be purchased for <\$200 (Adv. Mater. Technol. 2019, 4, 1900503). No toxic or expensive chemicals are involved.

Q2. The optical encoding demonstrated in Figure 5a is not unclonable, because it uses the optical pattern (spot detection) for anticounterfeiting. Although each spot is unclonable (size, shape, etc.),

the whole optical pattern is not unclonable if just detecting the presence of spot. Because the pattern is fabricated by the deterministic laser printing process, which could be easily copied by counterfeiters if they know the optical patterns.

R: The optical pattern alone would be clonable, if only a simple spot presence detection (yes/no) would be considered. While laser printing is a deterministic process, the final optical pattern is decided by the random partial removal of the polymer spots. This is a stochastic process and cannot be easily copied by counterfeiters. We have now also updated the scheme for the synthesis process in Fig. S2 (see Fig. R1-1) with more details to clarify this. Yet, most importantly, we show the verification of the optical pattern with a machine learning algorithm, which can achieve pixel-wise dense matches at a fine level. Not only the presence but also the morphology of the spot is considered during identification. More details on this algorithm have also been included in the revised manuscript. Furthermore, the electric PUF is also influenced by the optical pattern, making it more difficult for a counterfeiter, even when just using a simple spot presence approach.

Fig. R1-1 (Figure S2) Synthesis process of a PUF device. A polymer spot array is generated on a glass substrate by LIFT. During the first spin-coating step, an iron nitrite solution randomly washes away some or parts of the polymer spots. After annealing at 500 °C for 10 min, a hematite nanofilm with micro-hole patterns is obtained. This (inhomogeneous) hematite layer functions as a laser absorber to carbonize the polymer films on top of it under laser irradiation. Then, the whole area is brushed with colloidal graphite, which is washed away together with the uncarbonized polymer.

Q3. The electronic resistance encoding in this paper is just 8-digit (or 32-digit) long hexadecimal coding, which is still relatively lower than some hexadecimal optical coding with 50 × 50 digit long (or larger) within only hundreds of micrometers detection range. Therefore, the capacity of resistance encoding should be improved.

R: We can print the electrodes in a much smaller size to prepare more electrodes for higher encoding capacities. The size of our electrodes for both 8- and 32-digit electronics is in the range of mm². Since our printing technology can easily fabricate devices in the range of μm² (Adv. Mater. 34 (8), 2108493; Adv. Mater. 34 (23), 2200359), hundreds of electrodes may be printed in one electronic device. However, it will be much more difficult to read these PUFs manually with an ohmmeter and may even require some professional equipment such as a semiconductor parameter analyzer. This is not beneficial to the main idea of the manuscript. We agree that optical channels have the advantage of providing extraordinary encoding capacities. Therefore, in our work, we use a combination of the

optical channel and the resistance channel. These two independent channels can be more efficient if they have complementary characteristics.

Meanwhile, with the development of high-resolution surface replication and nanomolding techniques, an optical channel alone has an increased risk of being counterfeited. The resistance channel proposed in our work is generated by the combination and random distribution of the conductive and insulating materials in the PUF device. Therefore, it is more resistant to surface replication attacks than the optical channel alone. In addition, to the best of our knowledge, optical channels are typically designed as binary codes. PUF devices using 16 different fluorescent tags with sharp and non-overlapping emissions are not realistic. By defining different gray levels, hexadecimal optical encoding could be possible, but the readout and authentication process would be significantly affected, seriously compromising the reliability of the PUF devices. The tolerance to surface replication attacks and the hexadecimal encoding ensure that the resistance channel is a secure channel while still being manually readable with a simple ohmmeter.

Q4. It seems time-consuming and inconvenient to conduct electronic resistance measurement for encoding. As it needs to connect the electrodes of ohmmeter with each specific electrode in such small PUF labels to collect each datapoint, which means that at least 8 measurements are needed for one label. Therefore, the time consumption, difficulty of operation, misoperation, etc. should be considered for practical applications.

R: The difficulty of the operation was considered when we designed the electronics. We presented electronics with 32 electrodes because the size of the electrodes is still significantly larger than the tips of common ohmmeter probes. We could easily prepare 100 times smaller electrodes by our printing technology, but this would be incompatible with the manual readout using ohmmeters. For our manuscript, we designed a different scenario. If we take 3x3 PUF electrodes as an example, the first readout of an inexperienced user could take up to two or three minutes. For the second and third readout and with some experience, the measurement time was significantly reduced to less than one minute. Yet, to fully address this concern, we designed and built a semi-automatic resistance measurement setup (Fig. R1-1). It allows simple positioning, electrode contacting, and rapid measurement with a rotary selector switch. Due to its modular concept, it can be easily adapted to other electrode configurations. A discussion has been added to the supplementary information (Fig. S14) in the revised version of the manuscript.

Fig. R1-1 Semi-automatic resistance measurement setup. Nine metal pins are arranged in a 3 x 3 rectangle to fit the contact electrodes of our standard PUF devices. The pins are spring-loaded and a hand gear allows them to be moved up and down for quick contact. A window in the table, directly below the pins with an LED, allows for easy repositioning and alignment of the PUF on the table. A rotary selector switch allows manual selection of the circuit, which is connected to the ohmmeter.

Q5. The PUF encoding area/spot is relatively large, which is easy to be copied compared with nano- or micro- PUFs.

R: We agree that nano-PUFs will be more difficult to copy, but they are not easily accessible to general end-users, which is a critical challenge in the field. The main goal we are pursuing in this project is end-user accessible PUF devices, that could be read by consumer devices. Nano-PUFs undoubtedly promise a higher level of complexity and security, but readout is time and cost intensive, requiring expensive dedicated equipment such as scanning electron microscopy (SEM), atomic force microscopy (AFM), and/or confocal Raman microscopy. These systems are not accessible to the general public and this was the challenge we tried to address with this work. The resistance channel proposed in our work is generated by the connection and random distribution of the conductive, low-conductive, and insulating materials in the PUF area. Therefore, it is more resistant to surface replication than typical optical nano-PUFs, and the hexadecimal encoding process also guarantees that the resistance channel is sufficiently secure.

In addition, the size of the optical pattern (spots and spot spacing) can be effectively reduced by employing advanced printing technologies (photolithography, two-photon polymerization, dip-pen nanolithography). The design and synthesis process of our approach should be easily adaptable to these printing technologies.

Q6. Figure 2f is not clear, the degree of darkness is representing the resistances variations? Or? And the other two (middle and right) figures of 2f are not mentioned in the main text.

R: Thank you for pointing this out. We have now updated the explanation in the caption of Figure 2 and in the main text of the revised manuscript.

The degree of darkness in Fig. 2f represents the resistance variation caused by a thickness gradient of the hematite layer (from left to right: light orange to dark orange). Since the hematite precursor is introduced by spin-coating, and the initially printed polystyrene spot pattern can also affect this spin-coating process, the resulting hematite nanolayer is inhomogeneous in its thickness, indicated by an orange color gradient. Thicker hematite will have a higher laser absorption, resulting in stronger carbonization of the polymer layer (darker). The middle picture in Fig. 2f shows an electronic pattern without a polystyrene spot pattern. As shown in Fig. 2a&b, this will already cause some randomness in the electrode resistance. The right image shows the final PUF electronics with significantly improved PUF properties, caused by both the local inhomogeneities of the hematite layer and the randomly sized micro-holes due to the polystyrene spot pattern.

Q7. How Figure 2a and 2b are obtained should be explained in Methods.

R: Thank you. The method section has been updated in the manuscript for Fig. 2a & b.

Measurement and design of Fig. 2a & b: For each of the eight electrodes of a PUF device, the resistance was measured three times, the average was calculated, and arranged in a 3x3 matrix. Then, the median resistance of the eight electrodes was calculated and placed in the center of the 3x3 matrix. Based on this 3x3 matrix, we generated a 2D contour map in OriginPro 2022. The display mode was set as a grayscale color palette without contour lines.

Q8. From the height map of the micro-holes (Figure 3c), three layers (insulating, slightly conductive, conductive) can be distinguished. However, in Figure 1f, the resistance of thin layer electrode has a large range. So, is the resistance map from the height map trustworthy in Figure 3c?

R: The electrical resistivities of soda lime glass, hematite, and carbon are 7.94×10^{20} - $7.94 \times 10^{21} \Omega \cdot \text{cm}$ (Ashby, M. F. in *Materials and the Environment*, 2013, 459-595), $\sim 10^6 \Omega \cdot \text{cm}$, (Dimopoulos, D. in *The Future of Semiconductor Oxides in Next-Generation Solar Cells*, 2018, 533-543), and $\sim 3.5 \times 10^{-3} \Omega \cdot \text{cm}$ (Serway, R. A., *College Physics*, 2003), respectively, which are across a large range. Due to the randomly distributed micro-holes, the connection wires between electrodes bypass different combinations of conducting (carbon on top of hematite) and insulating areas (bare glass). Therefore, the resistance of the electrodes shows a large range. The resistance map from Figure 3c is the proposed explanation based on the thickness step map. It should be reasonable considering the layer structure of the electronics. Nevertheless, it caused confusion and raised concerns. Thus, we revised the manuscript by replacing the resistance description with a general material height description in Figure 3c.

Q9 Some quantitative analysis of the height maps shown in Figure 5b and 5c should be conducted. As it is hard to tell from the figures that there are no significant differences before and after treatment.

R: We have analyzed the surface roughness (root-mean-square (RMS) height standard deviation) of the electrode areas before and after the treatments. Furthermore, we analyzed the average step heights at the edge of the electrodes, comparing the relative height of the electrode area with the surrounding hematite layer. The RMS roughness differs only very slightly (<5 nm), while the edge step height differed a bit more (~20 – 35 nm) in the selected areas. This might be also influenced by the slight differences in the required manual selection of the measurement areas.

Before heat treatment

After heat treatment

Electrode RMS roughness

108.8 nm

103.6 nm

Edge step height

305.8 nm

283.5 nm

VSI measurement area selection

Before tape treatment

After tape treatment

Electrode RMS roughness

109.9 nm

108.7 nm

Edge step height

236.6 nm

271.2 nm

VSI measurement area selection

Reviewer #2 (Remarks to the Author):

This manuscript reported the printed PUFs having both resistive and optical keys. Authors claimed that their dual key PUFs offer high complexity and data capacity, which can be easily read out using common equipment such as an ohmmeter and an optical microscope.

While I find their concept interesting, I have reservations about their main argument. The authors claim that the combination of resistivity and optical readouts results in significantly higher complexity and randomness. However, based on the provided data, the resistivity readout does not exhibit sufficient complexity as a PUF; in fact, it performs poorly.

The inter-device uniqueness for the resistivity readout, measured using an ohmmeter, is reported as 0.937 ± 0.071 . Although the authors insist that this value indicates high randomness, it deviates significantly from the ideal value of 0.5. Contrary to their claim, this result demonstrates that the resistivity readout offers such poor randomness that it is challenging to use it as a PUF.

R: Thank you for your interest in our work. We are afraid, but there is a misunderstanding regarding the inter-device uniqueness of the resistance readout. While it is true that the ideal value of inter-device uniqueness for binary codes/keys is 0.5, this is different for non-binary codes. The uniqueness between any two PUF patterns is generally defined as:

$$\text{uniqueness} = \frac{2}{k(k-1)} \sum_{i=1}^{k-1} \sum_{j=i+1}^k \frac{\text{HD}(R_i(n), R_j(n))}{n}$$

where $R_i(n)$ and $R_j(n)$ are the n -value responses of the i^{th} and j^{th} PUF patterns, respectively, and k is the total number of the PUF patterns.

Based on this equation, the ideal uniqueness is 1/2 for binary keys, 2/3 for ternary keys (Nat. Nanotech., 2016, 11, 559; Adv. Funct. Mater., 2023, 33, 2211762), ..., 15/16 for hexadecimal keys. The inter-device uniqueness for the resistivity readout is 0.937 ± 0.071 , which is close to the ideal value of $15/16 = 0.9375$, giving solid evidence for the high performance. We updated this discussion in the revised manuscript for the general readership to avoid any misunderstanding.

On the other hand, only the optical readout provides a reasonable inter-device uniqueness value (0.454 ± 0.057), which, however, is not even as high as that of numerous other previous optical PUFs. Furthermore, the authors argue that the resistivity range of 0 to 1500 k Ω can be divided into 16 levels, allowing for 16-bit operations. For this claim to hold true, all levels should be evenly addressed with the same probability. However, the resistivity measurements depicted in Figure 2c, d, e, and Figure 4a reveal a relatively ununiform distribution, with some resistance ranges (e.g., 1000-1500 k Ω) lacking any hits.

R: We apologize if we have not made the encoding process clear. Figure 2c and 2d are based on three samples, and each sample has its own y-axis to compare the general resistance distribution of the electrodes at different positions. The intention and display format of these figures was not to show the electrode resistance range. Figure 2e and Figure 4a are based on 16 samples to describe the encoding process. The non-uniform distribution is due to the small number of samples. When we increase the number of samples to 100, the inter-hamming distance of the devices fits well with the Gaussian distribution, which supports our claim of random distribution of the resistance. Moreover, the resistance range is not the same for all eight electrodes. The algorithm we developed for the encoding process analyzes the data for each PUF device independently and divides their specific ranges into 16 levels. This was necessary as the general resistance of PUF devices can be tuned to be more in the k Ω or more in the M Ω range.

Considering these points, the resistivity readout appears unsuitable for use in PUFs. While their printed device still functions as an optical PUF, I fail to see any significant excellence or advancement compared to previously developed optical PUFs.

Overall, I do not believe this work qualifies for publication in the renowned journal Nature Communications.

Minor comments:

1. Figure 5c displays resistance in the $M\Omega$ scale, while Figure 5b shows resistance in the $k\Omega$ scale. Is this a typographical error, or does the resistance actually vary to such an extent from sample to sample?

R: It is not a typographical error. The resistance can be tuned by multiple parameters, giving a wide accessible range. These parameters include laser scanning speed (Fig. S5 and S6 in the submitted version), polymer type (Fig. S6-S8 in the submitted version), and laser carbonization repetitions (Fig. S9 in the submitted version). The stability test in Figure 5 was performed on two random samples from the parameter scanning. If we choose a fixed set of parameters, the resistances of the obtained electronics will fall into a similar range; if different levels of resistance are beneficial in some specific applications, our approach can be adjusted.

2. Since resistance varies with the extent of oxidation, I recommend conducting a stability test against oxidation for an extended period or performing an accelerated-oxidation experiment under controlled conditions.

R: If we understand this comment correctly, an oxidation stability test over a long period of time is expected. In our printing process, carbonization is achieved by laser heating under ambient conditions. In the submitted manuscript, we performed a stability test at 200 °C since this is high enough for general use conditions of anti-counterfeiting labels. To determine the stability under prolonged oxidative conditions, we placed a PUF device in an air oven at 500 °C for 5 min. The carbon layer disappeared while the hematite layer remained unchanged at 500 °C. This suggests that the PUF electronics are stable for general use and can be erased at higher temperatures and rewritten.

3. Figure S10: The x-axis sampling space of the left-right histogram differs. It should be the same for comparison purposes.

R: Both histograms are obtained from 8 samples for the inter-correlation, and each sample is measured three times for intra-correlation. The histogram display is automatically optimized by the algorithm to better fit the Gaussian distribution. Therefore, the x-axis sampling of the figures were specifically optimized, which we believe is advantageous for comparison.

4. Figure 3: How did you obtain the height information from the conventional optical image shown in Figure 3a, and how did you identify the insulating, slightly conductive, and conductive regions based on that information? I couldn't find reasonable explanations in the manuscript or the supplementary information. Additionally, what is the conductivity range for each of the three regions?

R: The results from Fig. 3a-c are acquired by a white light interferometer which includes both bright field (Fig. 3a, b) and interferometry (Fig. 3c) options. Fig. 3c is the colormap from the interference field. It has been further analyzed by the supporting software MountainsMap 8.0 with automatic

threshold detection and the step method. This suggested the three-layer definition in Fig. 3c. The resistance distribution was derived from the combination of this layer structure and the known material properties. The reported electrical resistivities of soda lime glass, hematite, and carbon are 7.94×10^{20} - $7.94 \times 10^{21} \Omega \cdot \text{cm}$ (Ashby, M. F. in Materials and the Environment, 2013, 459-595), $\sim 10^6 \Omega \cdot \text{cm}$, (Dimopoulos, D. in The Future of Semiconductor Oxides in Next-Generation Solar Cells, 2018, 533-543), and $\sim 3.5 \times 10^{-3} \Omega \cdot \text{cm}$ (Serway, R. A., College Physics, 2003), respectively. However, since this Fig. 3c has also caused confusion with others, we have updated the detailed methods and discussion in the revised manuscript and now only include the material height without the resistance hypothesis.

5. Please provide the electrical conductivities for the pure carbonized-layer at 45° and 90° angles.

R: Pure carbonized layers without micro-holes have been generated with the same parameters as we used for the PUF electronics. The specific laser parameters for carbonization were 100 % power and 300 mm/s speed. The electrical resistivities of the pure carbonized layers at 0°, 45° and 90° angles have been measured and calculated as 3.82 ± 0.11 , 8.96 ± 0.46 , $9.81 \pm 0.40 \text{ M}\Omega \text{ cm}$, respectively.

6. Please include XPS and Raman data for the carbonized-layer and provide an explanation in the text.

R: Considering the chemical structure of the polystyrene (Fig. R2-1a), it only has sp^2 and sp^3 carbon without functional groups. It may not be helpful to distinguish polystyrene from amorphous carbon by XPS spectra. We prepared three samples for both XPS and Raman measurements. One sample with the standard parameters for the PUF electronics (100 % laser power and 300 mm/s writing speed), labelled as 'High laser power'. The second sample was prepared with 80 % laser power and 300 mm/s writing speed, labelled as 'Low laser power'. The third sample was prepared without the carbonization step, labelled as 'PS control'. Peaks for sp^2 and sp^3 carbon can be observed in the C 1s spectra of the three samples. The ratio between these two peaks falls between 0.38–0.44. Since XPS only analyzes the first 10 nm of the surfaces, which is prone to contamination from the environment, we would not suggest interpreting too much into the slight difference of this ratio. Different surface treatments have been considered including oxygen plasma and Argon cluster etching. However, they will chemically change the surface of the thin polystyrene layer. We also performed Raman analysis of the three samples (Fig. R2-1c). A strong signal from the glass substrate dominates the spectra which interferes with the observation of the materials on the top.

Fig. R2-1 (a) Chemical structure of polystyrene. (b) XPS C 1s and (c) Raman spectra of the three samples: PS control, Low laser power (80 % laser power, 300 mm/s writing speed), and High laser power (standard parameter: 100 % laser power, 300 mm/s writing speed).

7. It would be helpful to label the electrode numbers in Figure 2f to aid reader understanding.

R: Thank you for this suggestion. The electrode numbers have now been added to Fig. 2f.

8. Hematite is an optically interesting material with a conductivity of approximately 10-11 S/cm at room temperature. If the electrical conductivity of hematite is improved, what is the expected performance of this operation?

R: The electrical conductivity of the hematite is not changed, but we introduce a conductive carbon layer on top of the hematite. The hematite simply functions as an absorber, which drastically heats up upon laser irradiation and causes the top polymer thin film coating to carbonize in the desired irradiation pattern. We did not expect any significant change of conductivity of the hematite itself. As shown in the extended oxidation experiment (5 min @ 500 °C), the carbon can be completely removed, without any change to the hematite.

9. In Figure 2g, the image of the carbonized-layer exhibits periodic light and dark patterns indicating

different conductivities. This could have a significant impact on ohmmeter characterization and may negatively affect reproducibility. Could the authors explain the rationale behind this approach?

R: Fig. 2g of the submitted manuscript was not an experimental result but the screen snip from the supporting software for the laser printing system. The periodic light and dark patterns were caused by moiré patterns due to display issues (size reduction of high-resolution lines). We apologize for the misleading figure and have now replaced it with a new scheme in the revised manuscript.

10. Contrary to the description on Page 5, line 144, where electrodes #1, #3, #5, and #7 in diagonal positions are said to have high resistance, the resistance of #3 electrode does not appear to be large. Could the authors provide clarification on this inconsistency?

R: Thank you for this comment and we agree. The resistance is not fully determined by the configuration of the electrodes and connection wires but also by the underlying hematite layer. We have updated the sentence with a more consistent version:

'The resistance of E_w devices was typically higher for electrodes at the diagonal positions (Fig. 2b,d, electrode #1) and lower for those in the horizontal direction (Fig. 2b,d, electrode #4).'

Reviewer #3 (Remarks to the Author):

This work proposes a method to create a PUF for an anti-counterfeiting purpose. While its novelty requires justification (its novelty is ambiguous so far), the method itself is technically sound. Below concerns need to be taken into account. I recommend a major revision to address the following points.

1. Firstly, this approach is suitable for device identification, not authentication. The authors can gain a detailed understanding between PUF identification and authentication by referring to [8]. Re-read [8] very carefully.

R: We re-read reference [8] thoroughly and also found the definition of authentication in a book from the perspective of computer science [Maes, R. (2013). PUF-Based Entity Identification and Authentication. In: Physically Unclonable Functions. Springer, Berlin, Heidelberg]. After discussions, we agree identification is more precise for the scenario. Thank you for pointing this out. We have revised the manuscript and SI by replacing 'authentication' with 'identification'.

2. Using multiple characteristics e.g., both resistance and optical parameters concurrently to create a PUF has been investigated in the literature. There is a lack of comparison with existing closely-related works. Performing this comparison thoroughly would help to support novelty of your work.

R: Chemistry and materials science have a rich portfolio of characterization methods. Multiple characteristics from different channels are a highly efficient strategy to improve the security level of PUF devices, especially for those fabricated by chemical methods. As suggested, we have now compared our work with the recent literature on chemical PUF devices, characterized with multiple methods, shown in Table R3-1. Chemical PUFs generally have a large challenge-response pair space, but their practicality is hindered by expensive and bulky equipment for characterization. The equipment is not only expensive but often requires skilled personnel and time-consuming readout procedures. Our proposed PUF device can be characterized using a household ohmmeter and a simple optical microscope without multiple lenses or filters. Importantly, these end-user friendly readout methods do not compromise the complexity of the PUF devices. The combination of two independent characterizations and the hexadecimal encoding process contribute to the high security level of the PUF devices. In conclusion, our proposed PUF (1) addresses the main challenge for chemical-based PUFs by offering characterization methods compatible with household devices; (2) provides greater complexity than the currently reported chemical PUFs. Thank you for the suggestion. We have now updated the table in the revised manuscript and supplementary information.

Table R3-1 Comparison between the proposed PUF device and the reported PUFs from the state-of-the-art.

PUF Devices	Key generation	Equipment	Printing ability	String length	Deep learning-based identification	Reference
nanoPUF	optical dichroism; resistance; Raman	Semiconductor parameter analyzer; polarized optical microscope; dispersive Raman spectrometer	Non-printable	2	No	Nat Electron 5, 433–442 (2022)
fractalPUF	Optical pattern; Raman (conceptual presentation)	Optical microscope; Raman spectrometer	printable	2	Yes	Nat Commun 14, 2185 (2023)
Carbon dots PUF	Fluorescence pattern; thickness	Fluorescence microscope; white-light interferometry	printable	2	Yes	Nat. Nanotechnol. 18, 1027–1035 (2023)
laserPUF	Optical pattern	Optical microscope	printable	3	No	Adv. Funct. Mater. 33, 2211762 (2023)
rPUF	Optical pattern, resistance	Ohmmeter; optical microscope	printable	16	Yes	This work

3. Line 22 does not seem to be appropriate for the abstract, as all the later descriptions have nothing with securing the AI.

R: The related sentence has been replaced by the description of the counterfeiting problem. Specifically, we added the following sentence to start the abstract in the revised manuscript:

‘Counterfeiting has become a serious global problem, causing worldwide losses and disrupting the normal order of society.’

4. Line25 ‘2-bit strings’ is an incorrect claim without specific condition. For example, SRAM PUF can indeed generate hundreds of thousands of bits.

R: The specific condition for “chemical-based PUF devices” has been added to make the description more precise and specified it to binary (= bit) keys, not 2-bit strings:

‘Physical unclonable functions (PUFs) are promising hardware-based cryptographic primitives, especially those generated by chemical processes showing a massive challenge-response pair space. However, current chemical-based PUF devices typically require complex fabrication processes or sophisticated characterization methods with only binary keys, limiting their practical applications and security properties.’

5. Line55-58. The comments about electronics PUF’s practicality is incorrect. Note that PUF has been commercialized and its main type PUF is SRAM PUF, which is indeed practical. This SRAM PUF does not need any extra hardware for authentication.

R: SRAM PUFs have shown a lot of advantages, but they are unlikely to end the PUF security race. Memory-based PUFs are not applicable to public CRP-based authentication since they are generally limited by the size of CRP space. It is necessary to develop other PUF devices including chemical-based PUFs. The whole paragraph discusses different types of chemical PUFs. By “electronic-based PUFs”, we actually mean those PUFs that are based on electrochemical properties, not silicon-based PUFs. Since it is misleading, we have revised the manuscript and replaced “electronic-based PUFs” with “electrochemical PUFs”.

6. Line 198, usually intra-distance (equals to bit error rate given digitized PUF response) should be approaching to 0 not 1 to indicate high reliability.

R: Thank you for pointing this out. We revised “intra-distance” into “intra-correlation”.

7. Line 201, is this identification capability is estimated by a 16-bit response? Usually, with same inter-distance (randomness) and intra-distance (reliability), increasing the PUF response length can improve the identification capability.

R: It is the analysis of the optical binary key. The hexadecimal resistance key has also been analyzed and is shown in Figure 4 in the manuscript.

8. Figure 5 a, what if the PUF is cut out and placed on the other painting’s frame to perform substitution attack?

R: Figure 5a shows a simplified use case, attaching the PUF to the frame of a picture for visualization purposes. A more realistic application for valuable products (e.g. picture) would likely involve the attachment of the PUF to the back of a picture, using a stable glue, which cannot be simply removed or cut off without destroying the picture (e.g., only with a specific solvent that would damage paint, canvas, or electronics). Moreover, we discussed with the technician from our glass workshop and learned that two pieces of glass can be connected seamlessly, to include the PUF into the cover glass of a product. Another possibility would be to use thinner glass as a substrate for the PUF, which breaks upon mechanical tampering. However, the specific PUF design and attachment will strongly depend on the use case. Attaching a PUF device to an already existing object often enables this kind of substitution attack by removing/cutting off the PUF and placing it onto another object, even when using soft optical PUFs. This disadvantage may in some cases also be an advantage (e.g., facile attachment), but may require additional security measures, as discussed before.

9. Reliability evaluations usually need to be performed under varying operating conditions. What if the temperature (e.g., room temperature) of the reference/enrolled response is different from the temperature (e.g., 0) of the regenerated response?

R: We have measured the resistance of electrodes before and after thermal treatment at 200 °C for 1 h. No significant changes were observed.

Fig. R3-1 (Figure 5c in manuscript). Thermal stability (200 °C, 1 h) test of the device. The surface topography and resistances are recorded before and after the test.

The measurement was performed after cooling down the device to room temperature (RT). In the case of measuring directly at higher or lower temperatures than the RT, the resistance value will undoubtedly change due to the temperature dependence of resistivity. This can be easily overcome by recording the resistance values at multiple temperatures for practical applications.

10. Can the optical PUF property be characterized with inter-HD and intra-HD in a similar way to Fig4?

R: Yes, the inter-HD and intra-HD of the optical PUF property can be found in Fig. S12 in supplementary information.

Fig. R3-2 (Figure S12 in SI). (a) Device uniqueness of the micro-hole patterns were characterized by inter-device Hamming distance (HD). The readout reproducibility of the micro-hole patterns was characterized by the intra-device HD, where each PUF pattern was scanned twice. (b) Cumulative distribution functions, showing the probabilities of false authentication (FA) and authentication error (AE) as a function of decision threshold.

11. The adoption of a machine learning model needs more elaboration to detail its role and specific operation.

R: Since this algorithm has already been published and we only use it as a tool, we refer to the publication for further details. We have described the general working principle of the machine learning algorithm in the Methods section as follows:

***LoFTR algorithm.* The LoFTR algorithm (Jiaming Sun, Zehong Shen, Yuang Wang, Hujun Bao, Xiaowei Zhou. LoFTR: Detector-Free Local Feature Matching with Transformers. CVPR, 00680v00681 (2021)) is a framework for image feature matching. Instead of sequentially detecting, describing, and matching image features, it first establishes a pixel-wise dense match and then refines the matches. Unlike traditional methods that use a cost volume to search for corresponding matches, the framework applies self- and cross-attention layers from its transformer model to obtain feature descriptors on both images. The global receptive field provided by the transformer model enables the LoFTR algorithm to produce dense matches, even in low-texture areas, where traditional feature detectors typically fail to produce repeatable points of interest. In addition, the framework model is pre-trained on indoor and outdoor datasets to recognize the type of image being analyzed, with features such as self-attention. The LoFTR module uses self- and cross-attention layers in transformers to transform the local features to be context- and position-dependent, which is crucial for LoFTR to obtain high-quality matches on indistinctive regions with low-texture or repetitive patterns. For image pair (I^A, I^B) , their similarity is defined by the number of matched features:**

$$\text{similarity}(I^A, I^B) = \frac{N_{\text{matches}}(I^A, I^B)}{N_{\text{features}}},$$

where $N_{\text{matches}}(I^A, I^B)$ is the number of matching features and N_{features} is the number of extracted features. The LoFTR algorithm extracts 4800 features in total, however, for the border area, a mask has been applied for practical analysis. Therefore, the maximum number of possible extracted features is 4256. The resulting colormap shows the matching probability P_c . In the case of dual-softmax, it is obtained by

$$P_c(i, j) = \text{softmax}(S(i, \cdot))_j \cdot \text{softmax}(S(\cdot, j))_i .$$

The score matrix S between the transformed features is calculated by

$$S(i, j) = \frac{1}{\tau} \cdot \langle \tilde{F}_{\text{tr}}^A(i) \tilde{F}_{\text{tr}}^B(j) \rangle ,$$

where \tilde{F}_{tr}^A and \tilde{F}_{tr}^B are the added features, which enter the LoFTR module for processing.

12. Please carefully proof-read the manuscript for English language and expression. For example, the word “facile” is not a good word for a scientific paper. It is one of those strange English words that can both be a compliment and an insult depending on how it is said.....so is best avoided in a journal article.

R: Thank you for the suggestions. ‘Facile’ is actually frequently used for chemical processes in literature. We have now revised ‘facile’ to ‘efficient’ for a general readership. We have also asked help from a scientific writer from the group to improve the expression.

13. All single letter math variable like n must be in italic. All multiple letter variables, labels, and functions must NOT be in italic, For example, in your paper: softmax, matches, HD, ...etc must not be in italic.

R: Thank you for this professional suggestion. Single letter math variables have been revised to italics, multi letter variables are non-italics.

14. In math, when you have multiple letter names, do no mix cases. It does not look good. So for example: Similarity  similarity

R: We have replaced all capital letters in multi letter variables with small letters.

REVIEWERS' COMMENTS

Reviewer #1 (Remarks to the Author):

Thank you for providing clear and detailed responses to our questions. However, we still have some concerns and advice that we would like to address:

1. After reading the response to Q1, we have gained a better understanding of why you regard your fabrication process as facile. While it involves multiple steps, this is not uncommon in previous works. Additionally, earlier fabrication processes often required special conditions, such as high temperature, low yield, and evident toxicity, whereas the process in your work does not. To ensure clarity, we advise you to include a specific description in the manuscript about why your fabrication process is considered facile.

2. Upon reviewing your response to Q2, we feel that the manuscript may inadvertently misguide readers. In particular, the manuscript describes certain PUF characteristics of the optical pattern based on the position information of printed holes (e.g., Figure 3e). However, the position information of printed holes is not unclonable, as forgers can duplicate the label with the same hole positions through a deterministic process (such as laser printing). The optical pattern information becomes unclonable only with the fine-level recognition provided by the LoFTR algorithm. Consequently, analyzing the PUF characteristics of hole position information is not meaningful.

We recommend making the following changes in your manuscript: 1) Remove content related to PUF characteristics of hole position information; 2) Clearly state that fine-level recognition based on the LoFTR algorithm is a necessary condition for achieving unclonable optical pattern information.

We hope these suggestions will help improve the clarity and accuracy of your work.

Reviewer #3 (Remarks to the Author):

I am happy with the revision.

REVIEWER COMMENTS

Reviewer #1 (Remarks to the Author):

Thank you for providing clear and detailed responses to our questions. However, we still have some concerns and advice that we would like to address:

1. After reading the response to Q1, we have gained a better understanding of why you regard your fabrication process as facile. While it involves multiple steps, this is not uncommon in previous works. Additionally, earlier fabrication processes often required special conditions, such as high temperature, low yield, and evident toxicity, whereas the process in your work does not. To ensure clarity, we advise you to include a specific description in the manuscript about why your fabrication process is considered facile.

R: Thank you for this comment. To further clarify this, we have now included the following two sentences in the introduction and conclusion of the manuscript:

“Our process circumvents the need for sophisticated preparation conditions, toxic materials, and expensive noble metals.”

“Similar to other reported PUF device fabrication methods, our process involves several processing steps. However, they are simple and circumvent the problems in PUF material pre-fabrication that typically involve sophisticated synthesis conditions, toxic materials, or expensive noble metals.”

2. Upon reviewing your response to Q2, we feel that the manuscript may inadvertently misguide readers. In particular, the manuscript describes certain PUF characteristics of the optical pattern based on the position information of printed holes (e.g., Figure 3e). However, the position information of printed holes is not unclonable, as forgers can duplicate the label with the same hole positions through a deterministic process (such as laser printing). The optical pattern information becomes unclonable only with the fine-level recognition provided by the LoFTR algorithm. Consequently, analyzing the PUF characteristics of hole position information is not meaningful.

We recommend making the following changes in your manuscript: 1) Remove content related to PUF characteristics of hole position information; 2) Clearly state that fine-level recognition based on the LoFTR algorithm is a necessary condition for achieving unclonable optical pattern information.

R: We agree and have now removed Figure 3e and added the following statement: “A simple hole position analysis (e.g., hole position present/not present) would make the optical PUF label easily clonable, since counterfeiters could duplicate the pattern with the same hole positions through a deterministic printing process. To utilize the high randomness of the optical pattern, fine level recognition is required. Therefore, the open-source LoFTR...”

We hope these suggestions will help improve the clarity and accuracy of your work.

R: Thank you very much for your help in further clarifying these points for the general readership.